# The Influence of Abortion Law on the Frequency of Pregnancy Terminations—A Retrospective Comparative Study

**DOI:** 10.3390/ijerph18084099

**Published:** 2021-04-13

**Authors:** Kornelia Zaręba, Stanisław Wójtowicz, Jolanta Banasiewicz, Krzysztof Herman, Grzegorz Jakiel

**Affiliations:** 11st Department of Obstetrics and Gynecology, Centre of Postgraduate Medical Education, 01-004 Warsaw, Poland; paniehermanie@gmail.com (K.H.); grzegorz.jakiel1@o2.pl (G.J.); 2Department of Medical Psychology and Medical Communication, Medical University of Warsaw, 00-581 Warsaw, Poland; stwojt@o2.pl (S.W.); jolantabanasiewicz.wum@gmail.com (J.B.)

**Keywords:** pregnancy termination, abortion, abortion law

## Abstract

Abortion law is one of the main factors influencing the number of abortions performed in a country. The study aimed to assess the influence of abortion law on the number of performed terminations with particular attention paid to pregnancy terminations due to fetal defects. The retrospective comparative analysis of statistical data included on the governmental websites of Poland and the UK was performed. The average of 190,733.1 terminations were performed in the United Kingdom in the years 2009–2018 with the average of 2820.9 due to fetal defects. At the same time the average of 858.6 terminations were performed in Poland with 820.7 due to fetal defects. Population size is the only significant predictor of the number of terminations in the United Kingdom. The increase in the number of deliveries and population in Poland was not linked to the increase in the overall number of terminations or terminations due to fetal defects. It might be due to the unavailability of pregnancy terminations in many places. The radicalization of abortion law exerts no influence on the decrease in the number of terminations due to fetal indications. The liberalization of abortion law promotes the increase in the number of terminations due to social indications.

## 1. Introduction

Abortion law is one of the main factors influencing the number of abortions performed in a country [1].

In 1932 Poland was the second country worldwide (after the USSR) which introduced the right to perform a legal abortion if maternal health was threatened or if a pregnancy occurred due to an unlawful act (Art. 32 of the Criminal Code as of 1932). Legal regulations concerning pregnancy termination in Poland are specified in the 7 January 1993 Act about Family Planning, Protection of the Human Fetus, and Conditions for Pregnancy Termination (Journal of Acts no. 17; item 78; as amended). It is considered that abortion law restrictions were the side effect of the democratic transformation in Poland and the return to the Catholic Church [2]. According to Polish law it is acceptable to terminate a pregnancy following the indication of the doctor and the opinion of the pregnant woman in three instances: (1)The pregnancy poses a direct threat to maternal health or life;(2)Prenatal screening or other medical evidence indicate a high probability of severe and irreversible fetal anomaly or an incurable life-threatening disease;(3)There is a reason to believe that the pregnancy is the result of an unlawful act.

The second of the above-mentioned situations is the most common cause of pregnancy terminations in Poland. In such a case a termination may be performed until the moment when the fetus is able to live independently outside of the mother’s body. The majority of lawyers specified that according to Polish law the fetal age should not exceed 6 months (22 gestational weeks) [2]. The third indication, i.e., the possibility of terminating a pregnancy resulting from an unlawful act, is applicable in case of only several women annually. Such a termination may be performed until the 12th gestational week. However, the tardiness of court procedures in Poland makes it impossible to obtain the permission of the court in such a short period.

As regards the United Kingdom, abortion is permissible according to the Abortion Act as of 1967, as amended in 1990 [3]. It may be performed if two specialist physicians acting in good faith view a pregnancy as qualified for termination.

The following indications for a termination are listed in the document:(a)The pregnancy has not exceeded its twenty-fourth week and that the continuance of the pregnancy would involve risk, greater than if the pregnancy were terminated, of injury to the physical or mental health of the pregnant woman or any existing children of her family; or(b)The termination is necessary to prevent grave permanent injury to the physical or mental health of the pregnant woman; or(c)The continuance of the pregnancy would involve risk to the life of the pregnant woman, greater than if the pregnancy were terminated; or(d)There is a substantial risk that if the child were born it would suffer from such physical or mental abnormalities as to be seriously handicapped.

A selective termination may be performed until the 12th gestational week in the United Kingdom in case of the implantation of a higher number of embryos obtained as a result of IVF procedures. Over 83% of selective terminations were performed due to fetal defects [4]. 

The analysis of reports issued by the Department of Health, UK Government, showed that the average of 190,733.1 abortions were performed annually in the UK, mostly due to social indications (points a and b) [4,5,6,7,8,9,10,11,12,13]. At the same time the average of 858.6 terminations were performed in Poland almost only due to fetal defects [14,15,16,17,18,19,20,21,22]. 

Tightening up abortion law does not reduce the number of pregnancy terminations performed. It only reduces their safety. Women are made to undergo illegal terminations outside medical facilities. We are unable to provide the reliable assessment of the grey market in Poland. However, the number of illegal terminations is assessed at approximately 100,000 annually [23]. In order to seek help numerous women travel to other European countries such as Slovakia, Germany or the UK. They also order abortion pills via the internet. 

Termination is reimbursed with state funds both in Poland and in the UK. It is a technically simple procedure. A total of 98% of terminations in the UK are reimbursed by the NHS (National Health System) [11]. Professor Lesley Regan from Imperial College in London indicated that, regarding the average age of sexual initiation in the United Kingdom being 16 and the average age of conception at 30, women from the UK use contraception for 16 years [3]. The average age of sexual initiation is 17–18 years in Poland [23]. Regardless of abortion law women will always seek abortion possibilities. Currently, the Constitutional Tribunal in Poland banned abortion due to fetal defects. According to the justification it was unconstitutional. The statement issued by the Constitutional Tribunal invoked the necessity to protect human life from the moment of conception. Therefore, we decided to verify whether tightening up the abortion law influenced the number of pregnancy terminations, with particular attention paid to pregnancy terminations due to fetal defects.

## 2. Materials and Methods

### 2.1. Material

We decided to conduct the retrospective comparative study of two European countries for which the statistics concerning the number and indications for terminations were available: Poland which is a country with one of the most restrictive abortion laws and the United Kingdom with its liberal abortion law. The data were retrieved from reports available on governmental websites: the Department of Health and Social Care Abortion Statistics for England and Wales and the Report of the Council of Ministers of the Sejm of the Republic of Poland. We analyzed data concerning the years 2009–2018.

We compared available information concerning the number of terminations comprising fetal defects as an additional variable and compared them with the population size and parity of the respective country. It was impossible to assess the grey area of illegal abortions in Poland, so population size was additionally considered. The number of deliveries only reflects the number of pregnancies which were continued. 

The study was conducted in accordance with the Declaration of Helsinki.

### 2.2. Methodology

The comparison of data from both countries as regards the analyzed factors was performed with the t-Student test. The correlation between those factors was calculated with the Pearson correlation coefficient. The determination of the predictors of the number of terminations was performed with linear regression analysis. We assumed the lowest *p-*value at 0.05. Logistic regression analysis was performed. 

Due to a significant discrepancy between the population of Poland and the United Kingdom the following percentage indexes were developed in order to eliminate the difference related to the population size:The percentage of fetal defects in the overall number of terminations = (fetal defects/number of terminations) × 100.The percentage of terminations in the overall number of deliveries = (number of terminations/number of deliveries) ×100.The percentage of fetal defects in the population = (fetal defects/population size) × 100.The percentage of terminations in the population = (number of terminations/population size) × 100.The delivery rate in the population = (number of deliveries/population size) × 100.

The study was conducted with IBM SPSS Statistics for Windows, Version 24.0., Armonk, NY: IBM Corp. (Released 2016).

## 3. Results

In 2009–2018 significantly more terminations were performed in the UK compared to Poland (Table 1). The average of 190,733.1 terminations were performed in the United Kingdom in the years 2009–2018 with the average of 2820.9 due to fetal defects. At the same time the average of 858.6 terminations were performed in Poland with 820.7 due to fetal defects. An increasing trend in the number of terminations was observed both in Poland and in the United Kingdom.

Pregnancy terminations in Poland accounted for 0.22% of the average delivery rate. The respective percentage in the UK was 26.9%. The average number of terminations due to fetal defects compared to the number of delivered babies was 0.4% in the UK and 0.2% in Poland.

The Pearson correlation coefficient showed that the increase in population size was correlated with the overall number of terminations and the number of terminations due to fetal defects. However, no correlation was observed with the number of deliveries in Poland. The respective values were different in the United Kingdom (Table 2), where the reduction in the population size was associated with the increased delivery rate and the decreased number of terminations due to fetal defects. In Poland the increase in population size was associated with the decreased number of terminated defective fetuses and the overall number of terminations.

The increased number of deliveries was not associated with the number of terminations in general and due to fetal defects. However, a higher number of fetal defects was related to the increase in the number of terminations in Poland (Table 3).

The number of deliveries in the UK was negatively correlated with the number of terminations, number of terminations due to fetal defects and the frequency of trisomy. However, the correlation with the number of terminations was statistically insignificant. Fetal defects were positively correlated with the number of pregnancy terminations and the number of trisomy cases with both those correlations being statistically insignificant (Table 4). Therefore, population size was the only significant predictor of the number of terminations in the United Kingdom. The remaining variables were statistically insignificant and were not included in the model (Table 5).

As regards Poland, the number of defective fetuses was the predictor of the number of terminations (Table 6).

Comparative analysis revealed that fetal defects being the reason for terminations were much more common in Poland than in the UK. However, terminations were less frequent than in the UK. Significantly fewer fetal defects were diagnosed in Poland than in the UK. Furthermore, fewer deliveries were reported in Poland compared to the UK (Table 7).

## 4. Discussion

The increase in the number of terminations in Poland is associated with the development in the area of prenatal diagnostics and, what follows, the increased detectability of fetal defects. Furthermore, the social awareness of the possibility of undergoing abortion increased. However, even if the number was doubled, it increased by fewer than 500 terminations as regards all pregnancies in the country. An increase of over 1000 terminations due to fetal defects was also reported in the United Kingdom, which was probably related to analogous reasons. The overall number of abortions in the UK was increased by almost 10 thousand. According to the statistics the abortions were mostly performed due to social indications. During the analyzed period, i.e., in 2009–2018 the law did not change either in Poland or in the UK. Therefore, it may be assumed that the fluctuations in the number of performed abortions were to a larger extent due to socioeconomic causes in the UK. As regards terminations for medical indications—they were due to the progress in prenatal diagnostics in the UK and Poland. A high percentage of abortions in the UK compared to Poland was due to the extensive list of indications. Abortion is permissible if two specialist physicians acting in good faith view a pregnancy as qualified for termination according to the Abortion Act as of 1967 (as amended in 1990). Apart from conditions similar to the Polish ones, if delivering an offspring carries a higher risk of psychological complications in the mother and her other children than in case of pregnancy termination it is also considered an indication for a termination [12]. In Poland the risk of psychological complications in the mother and her other children does not constitute an indication for pregnancy terminations. However, actually, all social indications are associated with such a risk. 

In Poland the increase in the number of deliveries was not related to the increase in the number of terminations and fetal defects. Significantly fewer fetal defects were diagnosed in Poland than in the UK. Such a phenomenon might result from the regionalization of facilities offering prenatal diagnostics and pregnancy terminations. They are not performed in numerous provinces, because physicians invoke the conscience clause [24]. Pregnancy terminations due to fetal defects are performed on a larger scale in few centers in Poland. Comparative analysis revealed that fetal defects being the reason for terminations were significantly more common in Poland than in the UK, because abortion on demand was not performed. Terminations were also less commonly performed than in the UK, due to the limited list of indications. Moreover, fewer deliveries were reported in Poland compared to the UK which might be due to the lack of social support in the country or the possibility of terminating a pregnancy. 

Abortion on demand is mainly sought due to social indications, such as young age, no partner, no support in the family and society, and low income [1]. Currently, the support for young mothers in Poland, even those who have sick children, is negligible. Supporting a child without an additional source of income is not possible. It is one of the main reasons reported by women who decided to terminate a pregnancy [25]. A project entitled “Za życiem” (“Pro-life”) was developed in Poland in order to reduce the number of pregnancy terminations. It is addressed to mothers whose children were diagnosed with a severe and irreversible disability or an incurable life-threatening disease which developed prenatally or during delivery. Therefore, an act was passed on the 4th of November 2016 concerning the support of pregnant women and families “Pro-life” (Journal of Laws, item 1860, as amended). Subsequently, “Pro-life”—a program of comprehensive support of families was adopted [21]. The following tasks are included in the program: providing coordinated care to pregnant women with particular attention paid to complicated pregnancies (also comprising psychological care), prenatal diagnostics and therapy and palliative and hospice care. A one-time benefit is awarded after the birth of a live child with a severe and irreversible disability or an incurable life-threatening disease. Regrettably, the support offered by the state for the care of a sick child is minimal and the majority of mothers decide to terminate a pregnancy. Therefore, it seems more justified to increase financial and social support. It might contribute to the reduction in the number of abortions due to social indications and appropriate sex education which might limit the number of unwanted pregnancies. Currently, sex education is removed from schools in Poland. Introducing more restrictions into abortion law, the criminalization of abortion and the lack of sex education are a slippery slope to an increased number of unwanted pregnancies and, therefore, to illegal abortions.

A comparative study between abortion policies in France and the United Kingdom demonstrated that physicians and anti-abortion activists were the main elements of shaping the social awareness of abortion [26]. The Society for the Protection of Unborn Children was established in 1967 and it is now the oldest anti-abortion organization worldwide. It should be noted that the majority of anti-abortion activists have a religion-related motivation [27]. A study conducted in the UK showed that anti-abortion groups were characterized by a high level of conservatism and usually belonged to the Roman Catholic Church [27]. However, they more frequently invoked human rights than Catholic doctrine. Numerous organizations also invoke the claim that motherhood is the natural duty of women. However, according to statistics the number of individuals declaring the Catholic denomination decreased in the UK [28]. A similar situation is observed in Poland. Despite the fact that the Catholic denomination is commonly declared by the Polish, even Catholics practicing their religion are frequently in favor of abortion [29]. Our research showed that 60% of women undergoing pregnancy termination in Poland were practicing Catholics. Imposing religious views on individuals who declare no religious belief may be viewed as the breach of their autonomy. Anti-abortion organizations mainly focus on anti-abortion protests. It seems more justified to refocus their enthusiasm on the prevention, i.e., contraception, or involvement in helping the parents of sick children who decided to continue the pregnancy.

In May 2018 the citizens of democratic Ireland voted in a referendum and approved of the possibility of pregnancy termination due to medical indications. Until that time abortion had been possible only in cases of the direct threat to maternal health. Illegal abortion was punished with up to 14 years of imprisonment. Currently it is available without medical indications until 12 gestational weeks. Furthermore, it is recommended to obtain instant contraceptive advice in order to prevent another abortion [30]. The authors indicated the cooperation between healthcare professionals and non-governmental organizations as the most advisable direction of developing reforms. The possibility of voting in a referendum seems to be the best solution in countries with the most restrictive abortion laws. Invoking social conscience or imposing one’s religious beliefs and viewpoint on other people is a mistake. The example of Ireland is the best evidence how distinct the law may be from the needs and expectations of the inhabitants of the country [31]. However, the Irish government decided to ask about the inhabitants’ opinions in a referendum. In case of considerable discrepancies in the opinion of the public, a referendum is the best method of counteracting subsequent strikes and pickets. The country remained strongly Catholic, but it left the decision to undergo abortion to the conscience of the citizens. 

A comparative study of the health exceptions related to pregnancy terminations in the UK, Colombia and Mexico revealed that abortion law may be developed in a way which facilitated a broad access to abortion, e.g., in case of the UK, or in a way which made it practically impossible to undergo a termination, e.g., in Mexico [32]. The interpretation of the law is the decisive factor. In countries inhabited by a high percentage of conservative citizens the law may be interpreted to the disadvantage of women deciding to undergo a termination. In case of a threat to maternal health abortion is permissible in all those countries. In the UK the opinion of a physician stating that the pregnancy may constitute a threat to maternal mental health is sufficient to justify a termination. Therefore, socioeconomic indications may also be taken into consideration. Mental health is not included in the list of indications for pregnancy termination in Mexico and Colombia. An article by Berer and Hoggart presented national strategies and trends in anti-abortion changes. They concluded that introducing abortion into the Criminal Code enhanced anti-abortion activism and intensified the attempts to breach women’s autonomy [33]. A similar situation is currently observed in Poland. Thousands of Polish citizens took to the streets on the 23rd of May 2018 to fight for abortion. The widespread mobilization of citizens was sufficient to block the attempt to change abortion law. However, in 2020, in the middle of the COVID-19 pandemic the Constitutional Tribunal in Poland banned abortion due to fetal defects stating it was unconstitutional. It caused another wave of protests. Legally imposed restrictions do not change the number of abortions, but they lead to the increase in mortality resultant from illegal attempts to terminate a pregnancy. They also make women seek help in neighboring countries with less restrictive abortion law [24].

Lesley Regan mentioned that the fight for abortion rights lasted over 30 years in the United Kingdom [3]. The authors stated that the most important task of all countries involved reducing the number of abortions by increasing the awareness of contraceptive methods, providing sex education and safe access to abortion for women if contraception failed, or in case of a medical indication for termination. The Guttmacher Institute reported that the number of terminations decreased from 46 to 27 per 1000 women over the past 25 years. The main reason indicated was the increased availability of modern contraception, decriminalization of abortion law and the safety of procedures conducted by professional medical personnel [34]. The decriminalization of abortion seems to be the main direction as regards the improvement in the area. As long as abortion is a crime, it will destroy women who were made to terminate a pregnancy. It will also be dangerous from the perspective of physicians who might perform the procedure [33].

### 4.1. Limitation of the Study

When interpreting the results of analyses the small size of study groups should be viewed in comparison with the population size. Therefore, apart from significance in correlation techniques it is also important to consider the values of correlations. 

The study was limited to two countries because of the lack of access to the statistics of other European countries. 

### 4.2. Strenghts of the Study

The study is the first one to compare society-related factors, such as the population and parity, which may influence the statistics of abortions performed in countries with considerably different abortion laws. It emphasizes numerous significant issues, including state policy and legislation, and indicates the direction for further management of the situation in countries with increasing social unrest related to abortion law. 

## 5. Conclusions

The radicalization of abortion law exerts no influence on the decrease in the number of terminations due to fetal indications. We observed an increased number of pregnancy terminations due to fetal defects in Poland and in the UK, which was probably due to the rapid development of prenatal diagnostics and higher medical awareness in the patients. 

The increase in the number of deliveries and population in Poland were not linked to the increase in the overall number of terminations, or terminations due to fetal defects. It might be due to the unavailability of pregnancy terminations in many places.

The liberalization of abortion law promotes the increase in the number of terminations due to social indications. Raising the awareness of contraception and suitable sex education at school are the only methods of reducing the number. 

A referendum regarding the permissibility of abortion carried out in the citizens might facilitate the determination of the form of the law accepted by the majority of the society.

## Figures and Tables

**Table 1 ijerph-18-04099-t001:** The number of terminations in the United Kingdom and Poland in the years 2009–2018 regarding fetal defects as the indication for terminations.

Year	The Number of Terminations in Poland	The Number of Terminations in Poland Due to Fetal Defects	The Number of Terminations in the United Kingdom	The Number of Terminations in the United Kingdom Due to Fetal Defects
2018	1076	1050	205,295	3269
2017	1057	1035	197,533	3314
2016	1098	1042	190,406	3208
2015	1040	996	185,825	3213
2014	971	921	184,571	3099
2013	744	718	183,331	2732
2012	752	701	185,122	2692
2011	669	620	189,931	2307
2010	641	614	189,574	2290
2009	538	510	195,743	2085

**Table 2 ijerph-18-04099-t002:** The Pearson correlation coefficient for the number of deliveries and terminations in the UK and Poland compared to the population size.

		Number of Deliveries		Number of Terminations		Number of Terminations Due to Fetal Defects	
		UK	Poland	UK	Poland	UK	Poland
Population	r	−0.883	0.223	0.631	−0.951	0.944	−0.966
	*p*	0.002	0.563	0.069	0.000	0.0001	0.000
	N	9	9	9	9	9	9

**Table 3 ijerph-18-04099-t003:** The Pearson correlation coefficient for the number of deliveries, terminations and fetal defects in Poland.

		Number of Deliveries	Number of Terminations	Number of Terminations Due to Fetal Defects
Number of deliveries	r	1	−0.279	−0.249
	*p*		0.467	0.517
	N	9	9	9
Number of terminations	r	−0.279	1	0.997
	*p*	0.467		0.0001
	N	9	9	9
Fetal defects	r	−0.249	0.997	1
	*p*	0.517	0.0001	
	N	9	9	9

**Table 4 ijerph-18-04099-t004:** The Pearson correlation coefficient for the number of deliveries, terminations and fetal defects in the UK.

		Number of Deliveries	Number of Terminations	Number of Terminations Due to Fetal Defects	Number of Terminations Due to Trisomy 21
Number of deliveries	r	1	−0.450	−0.831	−0.807
	*p*		0.225	0.005	0.009
	N	9	9	9	9
Number of terminations	r	−0.450	1	0.384	0.240
	*p*	0.225		0.308	0.535
	N	9	9	9	9
Fetal defects	r	0.831	0.384	1	0.957
	*p*	0.005	0.308		0.000
	N	9	9	9	9

**Table 5 ijerph-18-04099-t005:** Linear regression analysis for the number of terminations and the studied factors in the United Kingdom.

Model	Standardized Coefficients	t	Significance	95.0% Confidence Interval for B	
	Beta			Lower Bound	Upper Bound
Constant		−3.702	0.021	−1,283,436.022	−183,302.120
Population	2.672	5.159	0.007	0.007	0.023
Number of deliveries	0.394	1.303	0.262	−0.084	0.232
Fetal defects	−0.983	−1.404	0.233	−51.057	16.763

**Table 6 ijerph-18-04099-t006:** Linear regression analysis for the number of terminations and the studied factors in Poland.

Model	Standardized Coefficients	t	Significance	95.0% Confidence Interval for B	
	Beta			Lower Bound	Upper Bound
Constant		−2.443	0.058	−55,243.187	1404.193
Population	0.184	2.462	0.057	0.000	0.001
Number of deliveries	−0.029	−1.447	0.208	−0.001	0.000
Fetal defects	1.168	15.557	0.0001	0.975	1.362

**Table 7 ijerph-18-04099-t007:** Comparison of the number of terminations, deliveries and fetal defects between Poland and the United Kingdom (Student’s *t* test).

	Country	N	Average	SD	t	*p*
The percentage of fetal defects in the overall number of terminations	Poland	9	95.47	1.78	157.435	0.0001
	UK	9	1.52	0.20		
The percentage of terminations in the overall number of deliveries	Poland	9	0.23	0.05	−37.960	0.0001
	UK	9	26.97	2.11		
The percentage of fetal defects in the population	Poland	9	1.00	0.04	−3.191	0.006
	UK	9	1.10	0.08		
The percentage of terminations in the population	Poland	9	0.00	0.00	−102.022	0.0001
	UK	9	0.29	0.01		
The percentage of deliveries in the populations	Poland	9	1.00	0.04	−3.191	0.006
	UK	9	1.10	0.08		

## Data Availability

Data supporting reported accessed on 29 January 2021.

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
