# Peer review of "The Influence of Abortion Law on the Frequency of Pregnancy Terminations—A Retrospective Comparative Study"

_ijerph, 2021, doi:10.3390/ijerph18084099_

Round 1
Reviewer 1 Report
Thank you for opportunity to review the presented paper.
The article is clear, well written.
The title consistent with the problem actually presented.
The present study aimed at the comparative analysis of the influence of abortion law on the abortion rate. Particular attention was paid to terminations due to fetal defects which are currently banned in Poland. Please add in which countries the comparative analysis was performed.
The subject is topical and an important one.
Please add the full name of the NHS.
Please write who Lesley Regan is.
The interpretation of the results is clearly presented and adequately supported by the evidence..
The conclusions are logically valid and justified by the evidence adduced.
I read this article with pleasure.

Author Response
Dear Reviewer,
At the beginning, we want to thank you for all valuable remarks. We are happy that you enjoyed this topic. The manuscript has been revised according to your suggestions. Detailed answer is provided below.
Please add in which countries the comparative analysis was performed.
We have added the information according to your suggestions.
Please add the full name of the NHS.
We have added the information according to your suggestions.
Please write who Lesley Regan is.
We have added the information according to your suggestions.
Best regards,
Kornelia Zareba and Co-authors
Reviewer 2 Report
The article "The Influence of Abortion Law on the Frequency of Pregnancy 3 Terminations – A Retrospective Comparative Study", aims to think over pregnancy termination, considering the actual e former Polish legal framework. Authors compare data on pregnancy termination from nine years in the United Kingdom (UK) and in Poland.
The article is interesting, but some aspects are difficult to understand or are expected and eventually do not need a statistical demonstration.
The UK has a more liberal abortion law. Poland has a more conservative law. The main reason that allows abortion in Poland is fetus malformation. It is then expected that almost all the abortions in Poland are due to malformations (rape or sexual abuse and mother’s survival are residual in any country/legal framework). The statistics presented in the present study are redundant with the legal situation of each country. Eventually, the two countries' descriptive presentation would be more informative than the correlations presented.
English is not my natural language. Having that into consideration, I consider that the language could be improved, namely in simplifying very long phrases. Standardize the use of the acronym UK (United Kingdom) throughout the text.
I suggest focusing on the objectives of the article and organize the conclusions in accordance.
I send some general suggestions to the authors to improve the final manuscript.
Abstract
I suggest changing the Abstract, making a structured summary, including, a brief introduction to the theme/problem, objectives (not included), methodology, results, and conclusions.
Introduction
Line 26 to 29 – Objectives should be placed at the end of the introduction.
Line 29-30 - This information belongs to the methodology.
Line 41 – “…permission of the doctor…”. I think the doctor should give an indication of non-permission?
Line 49 – “…majority of authors”? What authors?
Line 73 to 76 - This information is repeated in the methodology, I think it should be presented in a more general way.
Line 85 – “ Lesley Regan….”. Place the reference number with the author
Line 85 to 87 - I think that they should include Poland's sex and reproductive health services as it is carrying out a comparative study.
Line 90 to 91 - The work's aim is repeated in several moments of the article, both the general-purpose, more "political", or the specific objectives of the paper, more pragmatic based on data.
Material and Methods
Line 94 to 96 – As already mentioned, the objectives of the study should be placed at the end of the introduction, not in the methodology.
The methodology is not clear, they must include the type of study, explain how the data were collected, systematize the information related to the data analysis (information is confused and repeated). Separate ethical issues from statistical data analysis.
Results
Please consider presenting the percentages rounded to the decimal or centesimal, not more than that. Eventually, the results could be clearer if expressed to one, ten, or hundred thousand to avoid the very low percentages.
Table1 - The title of the tables should be revised as it mentions percentages and the data in the table has no percentages.
Table 6 - I did not understand the importance of including the population variable as a predictor.
Table 7 - The title of the tables should be revised.
Line 170 to 172 - This information has already been referred to in the methodology.
Discussion
Line 185 to 186 – “The overall number of abortions in the UK was increased by almost 186 10 thousand (????). Both in Poland and in the UK the law did not change at that time ( ???) .” This information should be clarified
Line 195 to 196 – “ Such conditions may include: Edward’s syndrome, Patau syndrome,…”. This information is not related to the previous one, I do not think it is appropriate to make this connection.
Line 313 to 314 - “However, it demonstrated an important trend and the dangers associated with restrictions in abortion law. I do not think that the data allow us to make this statement.
Conclusion
I suggest focusing on the objectives of the article and organize the conclusions in accordance.
Good luck for your investigation
Author Response
Dear Reviewer,
At the beginning, we want to thank you for all valuable remarks. We are happy that you enjoyed this topic. The manuscript has been revised according to your suggestions. Detailed answer is provided below.
The statistics presented in the present study are redundant with the legal situation of each country. Eventually, the two countries' descriptive presentation would be more informative than the correlations presented.
There are no reliable research tools to determine the effect of the law on the number of pregnancy terminations. Therefore, we decided to compare the number of pregnancy terminations in individual countries comprising additional factors, such as parity and population size (it is the indirect form of the assessment of the grey area of illegal abortions in Poland). The results were characterized by several interesting trends included in the new conclusions.
English is not my natural language. Having that into consideration, I consider that the language could be improved, namely in simplifying very long phrases. Standardize the use of the acronym UK (United Kingdom) throughout the text.
We have corrected the text according to your suggestions.
I suggest focusing on the objectives of the article and organize the conclusions in accordance.
We have corrected the text according to your suggestions.
Abstract
I suggest changing the Abstract, making a structured summary, including, a brief introduction to the theme/problem, objectives (not included), methodology, results, and conclusions.
The Abstract has been changed according to your suggestions. According to the instructions for authors the abstract should be unstructured. It was not possible to fully describe the topic in the Abstract due to the word limit (200 words).
Introduction
Line 26 to 29 – Objectives should be placed at the end of the introduction.
We have corrected the text according to your suggestions.
Line 29-30 - This information belongs to the methodology.
We have corrected the text according to your suggestions.
Line 41 – “…permission of the doctor…”. I think the doctor should give an indication of non-permission?
We have corrected the text according to your suggestions.
Line 49 – “…majority of authors”? What authors?
It was changed into „lawyers”. The reference has been added.
Line 73 to 76 - This information is repeated in the methodology, I think it should be presented in a more general way.
We have corrected the text according to your suggestions.
Line 85 – “ Lesley Regan….”. Place the reference number with the author
We have added the information according to your suggestions.
Line 85 to 87 - I think that they should include Poland's sex and reproductive health services as it is carrying out a comparative study.
We have added the information according to your suggestions.
Line 90 to 91 - The work's aim is repeated in several moments of the article, both the general-purpose, more "political", or the specific objectives of the paper, more pragmatic based on data.
We have corrected the information according to your suggestions.
Material and Methods
Line 94 to 96 – As already mentioned, the objectives of the study should be placed at the end of the introduction, not in the methodology.
We have deleted the information according to your suggestions.
The methodology is not clear, they must include the type of study, explain how the data were collected, systematize the information related to the data analysis (information is confused and repeated). Separate ethical issues from statistical data analysis.
We have corrected the information according to your suggestions.
Results
Please consider presenting the percentages rounded to the decimal or centesimal, not more than that. Eventually, the results could be clearer if expressed to one, ten, or hundred thousand to avoid the very low percentages.
We have added the information according to your suggestions.
Table1 - The title of the tables should be revised as it mentions percentages and the data in the table has no percentages.
We have added the information according to your suggestions.
Table 6 - I did not understand the importance of including the population variable as a predictor.
It was impossible to assess the grey area of illegal abortions in Poland, so population size was additionally considered. The number of deliveries only reflects the number of pregnancies which were continued. The information has been added to the text.
Table 7 - The title of the tables should be revised.
We have added the information according to your suggestions.
Line 170 to 172 - This information has already been referred to in the methodology.
We have deleted the information according to your suggestions.
Discussion
Line 185 to 186 – “The overall number of abortions in the UK was increased by almost 186 10 thousand (????). Both in Poland and in the UK the law did not change at that time ( ???) .” This information should be clarified
We have added the information according to your suggestions.
Line 195 to 196 – “ Such conditions may include: Edward’s syndrome, Patau syndrome,…”. This information is not related to the previous one, I do not think it is appropriate to make this connection.
It is an editorial error. The information has been removed.
Line 313 to 314 - “However, it demonstrated an important trend and the dangers associated with restrictions in abortion law. I do not think that the data allow us to make this statement.
We have added the information according to your suggestions.
Best regards,
Kornelia Zareba and Co-authors
Round 2
Reviewer 2 Report
In general, the authors included the recommendations sent to improve the article. I send three small suggestions:
Line 10 and 26 - "Abortion law is the main factor ( ???) influencing the number of abortions performed in countries". I believe that you cannot affirm this, especially without using a reference with scientific evidence.
Line 95-96 - "No reliable research tools are available to determine the effect of the law on the number of pregnancy terminations." I would suggest removing this information.
Libe 184 - Table 7 - Please consider presenting the percentages rounded to the decimal or centesimal, not more than that. SD results with eleven decimal places don't make sense.
Very successful in your research!
Author Response
Dear Reviewer,
Thank you for all valuable remarks. The manuscript has been revised according to your suggestions. Detailed answer is provided below.
Line 10 and 26 - "Abortion law is the main factor ( ???) influencing the number of abortions performed in countries". I believe that you cannot affirm this, especially without using a reference with scientific evidence.
We have corrected the text to „Abortion law is one of the main factors” and added bibiography
Line 95-96 - "No reliable research tools are available to determine the effect of the law on the number of pregnancy terminations." I would suggest removing this information.
We have deleted the sentence according to your suggestions.
Line 184 - Table 7 - Please consider presenting the percentages rounded to the decimal or centesimal, not more than that. SD results with eleven decimal places don't make sense.
We have corrected the text according to your suggestions
Best regards,
Kornelia Zareba and Co-authors